# PET/CT for Predicting Occult Lymph Node Metastasis in Gastric Cancer

**Danyu Ma [1], Ying Zhang [1,2], Xiaoliang Shao [3], Chen Wu [1,2,*,†] and Jun Wu [1,*,†]**

1   Department of Oncology, The Third Affiliated Hospital of Soochow University, Changzhou 213003, China
2   Institute of Cell Therapy, Soochow University, Changzhou 213003, China
3   Department of Nuclear Medicine, The Third Affiliated Hospital of Soochow University, Changzhou 213003, China
*   Correspondence: chenwu@suda.edu.cn (C.W.); wujun683@czfph.com (J.W.)
†   These authors contributed equally to this work.

**Abstract:** A portion of gastric cancer patients with negative lymph node metastasis at an early stage eventually die from tumor recurrence or advanced metastasis. Occult lymph node metastasis (OLNM] is a potential risk factor for the recurrence and metastasis in these patients, and it is highly important for clinical prognosis. Positron emission tomography (PET)/computed tomography (CT) is used to assess lymph node metastasis in gastric cancer due to its advantages in anatomical and functional imaging and non-invasive nature. Among the major metabolic parameters of PET, the maximum standardized uptake value (SUVmax) is commonly used for examining lymph node status. However, SUVmax is susceptible to interference by a variety of factors. In recent years, the exploration of new PET metabolic parameters, new PET imaging agents and radiomics, has become an active research topic. This paper aims to explore the feasibility and predict the effectiveness of using PET/CT to detect OLNM. The current landscape and future trends of primary metabolic parameters and new imaging agents of PET are reviewed. For gastric cancer patients, the possibility to detect OLNM non-invasively will help guide surgeons to choose the appropriate lymph node dissection area, thereby reducing unnecessary dissections and providing more reasonable, personalized and comprehensive treatments.

**Keywords:** gastric cancer; lymph nodes; occult lymph node metastasis; PET/CT; maximum standardized uptake value (SUVmax); radiomics

## 1. Introduction

Gastric cancer is one of the most common diseases, ranking fifth in incidence and third in mortality worldwide in 2020. Specifically, it accounted for approximately 769,000 deaths, with at least 1 in 13 due to stomach cancer [1]. The diagnosis rate of early-stage gastric cancer is low, with more than half of the patients having advanced or metastatic disease at the time of diagnosis [2]. Currently, radical adequate surgical resection (R0) is the only curative therapeutic modality option for gastric cancer [3–5]. However, more than half of the patients experience recurrence after surgical resection [6]. Lymph node metastasis is considered a major risk factor for recurrence in patients receiving surgical resection [7]. Regional or distant lymph node metastasis presents in more than half of the gastric cancer patients at the time of initial diagnosis or surgery. Therefore, the diagnosis of positive lymph nodes is crucial for the staging, grading and prediction of survival of gastric cancer patients [8–11].

OLNM is defined as a clinical diagnosis of stage N0 that is not detected by conventional palpation and imaging (CT and PET/CT) and lymph node metastasis confirmed by postoperative pathological sections [12–15]. The incidence of OLNM in gastric cancer ranges from approximately 11% to 56% [16–19]. One study demonstrated that occult metastases were identified in approximately 16% of gastric cancer patients using $^{18}$F-FDG PET/CT [20–23]. Kudou [24] analyzed 117 patients with gastric cancer who were evaluated preoperatively with $^{18}$F-FDG PET/CT, and some of them had OLNM metastases detected by postoperative pathology. A study by Findlay [25] reported that $^{18}$F-FDG PET/CT was able to detect lymph node metastasis that was not detected by routine staging, and the gastric cancer patients who developed metastatic lymph nodes had a poorer prognosis. Other studies have consistently shown that patients with detectable OLNM had a significantly poorer prognosis [14,26]. These studies demonstrate the value of PET/CT as a more sensitive method to detect OLNM and facilitate prognosis prediction and recurrence prevention at an earlier stage [14].

Early detection of OLNM by PET/CT also benefits patients by improving the precision of surgical interventions. Lymph node dissection is an important part of surgical treatment for gastric cancer patients [27], and D2 resection is widely used in clinical practice [28]. However, unnecessary lymph node dissection is performed in about 85% of patients [29–31], which may lead to various surgical complications and reduced quality of life. If OLNM is detected before surgery, appropriate surgical procedures can be chosen and damage to patients can be reduced. Currently, only about 10% of gastric cancer patients receive PET/CT before surgery [22].

This article extensively reviews the current literature on the use of $^{18}$F-FDG PET/CT for detecting lymph node metastasis in patients with gastric cancer, as well as the development of new imaging agents and radiomics in this context. Current research progress, recent understanding and improvements and future prospects are discussed to comprehensively evaluate $^{18}$F-FDG PET/CT as a method enabling earlier and more precise treatments for gastric cancer patients.

## 2. Article Selection

We systematically searched PubMed/Medline and Web of Science databases for relevant articles from January 2012 to July 2022. We used the keywords: "gastric cancer", "occult lymph node metastasis", "lymph node metastasis", "PET/CT", "$^{18}$F-FDG", "metabolic parameters", "novel imaging agents" and "radiomics" to search English-based articles. In addition, we did not limit ourselves with gastric cancer research. We broadened our search by screening references of relevant studies for additional information that might be useful for our analysis. Approximately 423 relative articles were found. Among them, 125 articles were selected as references for this review article. The case reports, comments, perspectives, editorials, and research with unpublished results were not considered.

## 3. Review

1.     Advantages of 18F-FDG PET in detecting lymph node metastasis in gastric cancer

$^{18}$F-FDG PET is widely used in the evaluation of various tumors. In gastric cancer, it plays a special role in the staging of patients, monitoring of recurrence, and detection of tumor metastasis [32–35]. Lymph node size is an important indicator for lymph node metastasis assessment. A short diameter of $\geq$10 mm on CT images is considered indicative of metastatic lymph nodes [36,37], but the accuracy is limited by the size of the lymph nodes. More than 60% of metastatic lymph nodes in gastric cancer are smaller than 8 mm [38], resulting in a low detection rate by CT. Therefore, the absence of enlarged lymph nodes on CT is not equivalent to the absence of lymph node metastasis. The value of $^{18}$F-FDG PET/CT in addressing such insufficiency and detecting distant metastatic lymph nodes in patients with gastric cancer is discussed in previous studies. It has been suggested that $^{18}$F-FDG PET/CT relies on glucose metabolism rather than changes in lymph node size to diagnose lymph node metastasis [19,32,39], reflecting the metabolic information of tissues and cells at the molecular level. On PET images, positive lymph node metastasis is characterized by greater $^{18}$F-FDG

uptake in lymph nodes ≥ liver [40]. Besides, patients with [18]FDG-avid lymph nodes in the surgical area are at a higher risk of recurrence and death after surgery [41–43]. Previous evidence demonstrated that [18]F-FDG PET/CT was able to identify lymph node metastases that were not detected by routine staging, and these metastases were related to a worse prognosis [25]. Studies comparing [18]F-FDG PET/CT with CT demonstrated that [18]F-FDG PET/CT is superior in terms of OLNM identification and preoperative lymph node staging improvement [22,44]. Based on the evidence, [18]F-FDG PET/CT has the potential to become an ideal non-invasive imaging technique for the preoperative lymph node staging of gastric cancer.

2. Limitations and challenges of 18F-FDG PET in gastric cancer

Although the advantages of [18]F-FDG PET/CT in the staging of lymph node metastasis have been established, its diagnostic value of for lymph node metastasis in gastric cancer remains controversial [40,45,46]. For PET, some studies have reported high sensitivity and specificity of 79–85% and 87–92% [23,47–49]. For [18]F-FDG PET/CT, although high specificity at 73–92% has been reported for the diagnosis of lymph node metastasis in gastric cancer, the sensitivity was limited to only 40–54.7% [23,36,46,47]. These results reveal that several significant challenges need to be addressed before [18]F-FDG PET/CT can be applied more widely for predicting OLNM in gastric cancer:

(1) As the histological types in gastric cancer vary, the proportion of FDG-avid tumors accounts for only 60%, especially for those with non-intestinal tissue types (diffuse, mixed, and indolent cells) [22]. In addition, [18]F-FDG PET/CT is not sensitive to lymph node metastases from non-FDG tumor metastases [50–52].

(2) Some scholars believe that the size of metastatic lymph nodes is a critical factor in the evaluation of gastric cancer. Some of the metastatic lymph nodes may be smaller than 3 mm [19]. As this is lower than the spatial resolution limits of conventional PET scanners, PET/CT tends to miss some of the metastatic lymph nodes with smaller diameters [50]. This contradicts our view.

(3) High physiological uptake of [18]F-FDG by the normal gastric wall creates radioactive volume effects, and background noise, gastric peristalsis and the contraction of normal gastric folds can all hinder the detection of gastric cancer and LN metastases [51].

(4) Inflammation of the stomach, benign disease [32], and Helicobacter pylori infection can lead to aggregation of tracers, resulting in false positives [53–55].

All these factors may lead to a high false negative rate of [18]F-FDG PET/CT for the detection of OLNM in patients with gastric cancer. For this reason, there is a need to identify a highly sensitive, non-invasive imaging marker to predict OLNM.

3. Common methods for detecting OLNM based on [18]FDG

SUVmax is a metabolic parameter commonly used in [18]F-FDG PET/CT. It is a common parameter for assessing the status of lymph nodes and detecting the malignant behavior of the tumors. For example, SUVmax can be used to assess the response status of lymph nodes in the most regions with the most active glucose metabolism. Some studies pointed out that the greater the SUVmax, the higher the incidence of lymph node metastasis [56]. In Lin's study [56], 284 patients with NSCLC were divided into two groups with a SUVmax cutoff of 2.6, and it was found that the metastasis rate of OLNM in the SUVmax < 2.6 group was 1.0% (1/100), while that in the SUVmax ≥ 2.6 group was 12.5% (23/184). The study reported that the detection rate of OLNM gradually increased with higher SUVmax value. Similarly, Xu [57] et al. suggested that SUVmax > 9.7 was a predicter of OLNM in early-stage tongue squamous cell carcinoma. Higher SUVmax uptake values suggested the possible presence of OLNM [58].

For patients with gastric cancer, Kim et al. [50] demonstrated increased sensitivity of lymph node metastasis detection at higher T-SUVmax values. Consistently, Song et al. [59] showed that T-SUVmax reduced the false negative rate of lymph node metastases detection by [18]F-FDG PET/CT in gastric cancer patients, which indicates that the detection rate of OLNM was improved. Higher SUVmax of the primary tumor was associated with the occurrence of metastasis in OLNM and poorer prognosis. In addition, it has also been confirmed that the ratio of lymph node SUVmax to primary tumor SUVmax is more accurate than primary tumor SUVmax in predicting OLNM, with a sensitivity of 93–95% [60]. [18]F-FDG PET/CT has also been used to detect lymph node metastasis in gastric cancer in preoperative settings, and SUVmax was a significant independent predicter of OS (overall survival) and RFS (recurrence-free survival) in patients with preoperative lymph node involvement [61]. Therefore, the combination of SUVmax with histological staging and lymph node staging can potentially provide accurate treatment options [62].

However, SUVmax is not without drawbacks. Specifically, it is susceptible to patient's blood glucose, the timing of tracer uptake, respiratory motion, inter-scanner variability, image acquisition and reconstruction parameters, and inter-observer variability [63,64]. As a result, several studies have proposed SUVpeak, a hybrid SUV measurement method, to reduce noise [65] by including the local average SUV value of a group of voxels around the voxel with the highest activity [66]. The predictive value of SUVpeak for lymph node metastasis in patients with gastric cancer has been demonstrated by Oh [67].

SUV is still an important predictor for detecting lymph node metastasis. Although SUVmax is used most commonly, studies covering its application in predicting OLNM in gastric cancer patients are lacking, and there is a demand for more retrospective studies. Besides SUVmax, accumulating evidence in recent years has demonstrated the capability of [18]F-FDG PET/CT in detecting OLNM in various cancers using other common primary tumor metabolic parameters, including SUVmean, metabolic tumor volume (MTV), total lesion glycolysis (TLG), and standardized uptake ratio (SUR). Table 1 summarizes the calculation of different parameters. Nonetheless, such evidence has not been demonstrated in patients with gastric cancer. As relevant studies have demonstrated the correlation of these parameters with prognosis in gastric cancer patients, their prospects as imaging modalities to predict OLNM is still promising.

**Table 1.** Comparison of the calculation of different [18]F-FDG PET/CT parameters.

| | |
|---|---|
| SUV | SUV = activity concentration in tissue/activity per body weight injected [63,68]. Quantitative description of the glucose metabolism of the lesion. |
| SUVmax | SUVmax is the highest voxel value of focal uptake of the tracer in tumors and represents the most intensive [18]F-FDG uptake in tumors. |
| SUVmean | SUVmean is the average level of glucose metabolism. |
| SUVpeak | SUVpeak is the local average SUV value of a 1 $cm^3$ group of voxels centered on the hottest voxel point in the tumor [66]. |
| MTV | (i) Use SUVmax = 2.5 as the threshold to outline the volume, <br> (ii) Use 40% SUVmax as the threshold. It is the volume of tumor lesions with increased [18]F-FDG uptake within the established SUV range. |
| TLG | TLG = MTV × (tumor SUVmean/blood SUVmean). |
| HF | HF by linear regression analysis of the derivative (volume difference/threshold difference) of the SUVmax metabolic volume (V)-threshold (T) function. |
| SUR | SUR was derived from the tumor SUV to B-SUR, and the tumor SUV to L-SUR was derived. |

[18]F-FDG: [18]F-Fluoro-2-deoxy-d-glucose; SUV: Standardized uptake value; SUVmax: Maximum standardized uptake value; SUVmean: Mean standardized uptake value; SUVpeak: Peak-standardized uptake value; MTV: Metabolic tumor volume; TLG: Total lesion glycolysis; HF: Heterogeneity factor; SUR: Standardized uptake ratio; B-SUR: Blood SUVmean standardized uptake ratio; L-SUR: liver SUVmean standardized uptake ratio.

## 4. Potential Approaches for the Detection of OLNM in Gastric Cancer

*4.1. $^{18}$FDG-Based Methods*

### 4.1.1. MTV and TLG

MTV integrates tumor-related metabolic activity and tumor volume. In contrast to SUV, it does not describe the maximum or average glucose turnover rate at a specific point, but rather the glucose turnover rate for all lesions. Several studies [69–71] have shown that MTV is an independent risk factor, that higher MTV predicts OLNM, and that for patients with high metabolic parameters, surgical planning can be tailored to achieve the optimal treatment plan and improve patient prognosis based on PET results [72,73]. Xu et al. [69] showed that MTV was the only meaningful predicter of OLNM in both univariate and multifactorial analyses. The risk of developing OLNM increased at higher MTV values. Park et al. [70] analyzed 139 patients with small NSCLC to determine the predicters of $^{18}$F-FDG PET/CT for OLNM. In this study, they concluded that MTV could be used to predict OLNM, but caution is needed. MTV is currently widely used for detecting OLNM in patients with lung cancer, while there are insufficient data in gastric cancer, although its value in assessing the prognosis of gastric cancer patients has been demonstrated by emerging evidence [72,74]. MTV is different from SUVmax in that it is a volume-based PET metabolic parameter representing intact tumor biology instead of the single highest intra-tumor volatile activity. Although MTV has shown promise for predicting lymph node metastasis, SUVmax will likely remain the most widely used PET parameter, as the method of obtaining MTV has not been standardized and its application will require caution [75].

TLG is obtained by multiplying MTV by the SUVmean of $^{18}$F-FDG PET/CT. It is a combined parameter representing tumor metabolic volume and glucose metabolism levels. TLG has the potential to provide better diagnostic efficacy. When larger tumors undergo necrosis, MTV may decrease, but metabolism in non-necrotic areas is elevated, resulting in increased TLG. Ouyang et al. [76] retrospectively analysed 157 patients with stage cN0 lung adenocarcinoma and concluded that TLGsur (odds ratio, 1.024; *p* = 0.002) was the most potent associated factor for predicting OLNM in lung cancer patients. TLGsur was more favourable than other PET parameters for predicting OLNM in cN0 lung adenocarcinoma patients. However, although higher TLG often indicates a poor prognosis in patients with gastric cancer [73], further exploration is required as it is difficult to determine its effectiveness in predicting the OLNM of gastric patients with current methods.

### 4.1.2. HF

It has been found that the coefficient of variation of metastatic lymph nodes is significantly higher than that of inflammatory lymph nodes positive for FDG uptake [77]. In a recent report by Ouyang et al. [78], 215 patients with clinical T1-2N0M0 NSCLC squamous cell carcinoma (SQCC) were analysed. HF was obtained by taking 40% to 80% of the derivative of the SUVmax volume-threshold function. In multivariate analysis, only the HF of primary tumor was an independent predicter of OLNM in SQCC patients. HF was significantly associated with OLNM. Similarly, in a study by Kim et al. [79], HF was a predicter of regional lymph node metastasis in patients with esophageal cancer. HF has also been studied in the assessment of breast cancer, endometrial cancer and other tumors. Currently, there is no valid basis for using HF to predict OLNM in patients with gastric cancer.

### 4.1.3. SUR

As previously mentioned, SUV measurement is affected by a variety of factors that can lead to up to 30% variation. Some studies indicate that SUR can reduce the variability produced by SUV and ingestion time. It has been reported [76,80,81] that the SUR derived from the ratio of tumor SUV to normal tissue SUV has a better prognostic value than SUV. Shi et al. [82] retrospectively analyzed 124 patients with non-small cell lung cancer. In their study, the detection rate of OLNM was 15%, and they concluded that L-SURmax (standardized uptake value divided by liver SUVmean) of carcinoembryonic antigen and cytokeratin 19 fragment was effective in predicting OLNM. SUR overcomes the impact of tracer supply imaging techniques [80,83] and is more adequate than primary tumor SUV in tumour recurrence prediction and prognostic assessment [84,85]. However, SUR as an SUV-based parameter can be affected by the wide variations in SUV and the differences in tissue FDG distribution. Additional evidence is needed to clearly justify the potential of SUR as a replacement for SUV.

Through the comparison of various metabolic parameters based on $^{18}$F-FDG PET/CT, SUVmax remains dominant in predicting LNM metastasis in gastric cancer patients. Volumetric parameters such as MTV and TLG have shown better prediction of OLNM in patients with other tumors, and the improved diagnosis rate has been translated to more aggressive treatment modalities. The characteristics of commonly used metabolic parameters are presented in Table 2. Overall, the specific clinical value of volumetric parameters in OLNM prediction remains debatable due to the small sample size of existing studies and the more cumbersome measurement of TLG and MTV. Additional research and large-scale clinical trials are thus needed to further characterize the advantages and disadvantages of each. $^{18}$F-FDG PET/CT is a valuable imaging modality for the diagnosis of gastric cancer. However, as previously mentioned, the limitations leading to false-negative OLMN diagnosis in gastric cancer need to be accounted for first. The development of novel tracers will improve the sensitivity and specificity of OLNM diagnosis in gastric cancer to provide more personalized treatment.

**Table 2.** Summary of the characteristics of various $^{18}$F-FDG PET/CT parameters.

| Parameters | Summarize | Deficiency |
| --- | --- | --- |
| SUVmax | SUVmax is the most commonly used non-invasive metabolic parameter to predict tumor metastasis [40,86], SUVmax is now widely used for predicting OLNM in patients with lung cancer. The incidence of lymph node metastasis increased with higher SUVmax. | SUVmax only indicates a single voxel value and is susceptible to a variety of factors, such as blood glucose levels, inflammation, injection dose, imaging technical differences, etc. [63–65]. |
| MTV | MTV is the volume of tumor lesions above a certain metabolic threshold [87]. Some studies have demonstrated that MTV predicts survival prognosis better than SUVmax [88]. MTV has been proposed to be an independent prognostic factor of several cancers [75,89–91]. | The method of obtaining MTV is not yet standardized, and SUVmax is still the most commonly used parameter. |
| TLG | Some studies suggest that TLG may be superior to MTV and SUVmax [72,92]. TLG is a more accurate predicter of survival than MTV in lung, head and neck, gallbladder and soft tissue sarcomas [93–95]. It has the potential to become an important marker for predicting OLNM [96]. | The relationship between TLG and OLNM at the primary site of gastric cancer is still unclear. |
| HF | Some studies [79] have suggested that HF is an independent predicter of lymph node metastasis, and it has been applied to the evaluation of breast, oral, endometrial and other tumors [97,98]. | Tumoral metabolic heterogeneity is not well standardized and a feasible and highly reproducible method is needed to obtain heterogeneous parameters representing tumoral metabolic heterogeneity. |
| SUR | SUR is an SUV-based parameter that can be used as a potential alternative to SUV, complementing its limitations [80,81]. SURmax is another potential parameter for predicting OLNM. | SURs are usually derived from a region of interest (ROI) located within the aortic lumen, which is manually delineated in the CT image volume of a given PET/CT data. This manual delineation of ROI requires more care and time control and therefore creates additional workloads for the clinician [99]. |

SUV: Standardized uptake value; SUVmax: Maximum standardized uptake value; MTV: Metabolic total tumor volume; TLG: Total lesion glycolysis; HF: Heterogeneity factor; SUR: Standardized uptake ratio.

### 4.2. Novel Imaging Agents

#### 4.2.1. $^{68}$Ga-FAPI

Fibroblast activator proteins (FAP) are highly expressed in the stroma of many epithelial cancers. $^{68}$Ga-FAPI, a new target identified for tumour tracer development, has a high sensitivity for the diagnosis of primary tumours and metastases [100–103]. It is advantageous in that no diet or fasting is required. Moreover, it has better tumour-to-background contrast than $^{18}$F-FDG [104,105] and lower gastrointestinal uptake. Compared to $^{18}$F-FDG, $^{68}$Ga-FAPI provides superior detection of peritoneum, abdominal lymph nodes, and primary tumors and metastases in gastric cancer patients [106]. Pang et al. [107] performed a retrospective analysis of $^{68}$Ga-FAPI PET/CT in gastrointestinal tract tumors. The study reported a patient with gastric indolent cell carcinoma and found that OLNM was better detected using $^{68}$Ga-FAPI than $^{18}$F-FDG, which was subsequently confirmed by endoscopic US. The sensitivity of lymph node metastasis by pathology was stronger than $^{18}$F-FDG (79% vs. 54%, $p < 0.001$), but the specificity was not substantially higher. Another study used $^{68}$Ga-FAPI to detect primary gastric cancer and metastasis in 38 patients in a bicentric retrospective analysis [108]. One of the patients with lymph node metastasis ignored by $^{18}$F-FDG showed high uptake with $^{68}$Ga-FAPI, and postoperative pathology confirmed lymph node metastasis. This demonstrates the important role of $^{68}$Ga-FAPI in the detection of metastasis in gastric cancer. Although the results supported that $^{68}$Ga-FAPI was more sensitive than $^{18}$F-FDG for detecting lymph node metastasis in patients with gastric cancer, the small sample size did not adequately demonstrate that $^{68}$Ga-FAPI could enhance the detection rate of OLNM. Studies that included $^{68}$Ga-FAPI and $^{18}$F-FDG are summarized in Table 3.

**Table 3.** The clinical significance of using PET/CT to detect the OLNM.

| Author/Year | Types of Cancer | No. Patients | PET Imaging | Metabolic Parameters | No. of OLNM (%) | Sensitivity | Specificity |
|---|---|---|---|---|---|---|---|
| Hino [109]/2021 | Lung cancer | 598 | $^{18}$F-FDG | SUVmax | 17.06% | 88.40% | 41.80% |
| Pang [107]/2020 | Gastrointestinal tumors | 35 | $^{68}$Ga-FAPIs | SUVmax | 7.10% | 79.00% | 82.00% |
| Shi [82]/2020 | NSCLC | 124 | $^{18}$F-FDG | SUR | 15.00% | 94.70% | 57.10% |
| Xu [57]/2020 | Early-Stage Tongue Squamous Cell Carcinoma | 120 | $^{18}$F-FDG | SUVmax | 15.00% | 77.80% | 92.20% |
| Xu [69]/2019 | Esophageal squamous cell carcinoma | 84 | $^{18}$F-FDG | MTV | 46.03% | 51.20% | 83.70% |
| Ouyang [78]/2019 | NSCLC | 215 | $^{18}$F-FDG | HF | 16.70% | 88.90% | 61.10% |
| Ouyang [76]/2018 | Lung adenocarcinoma | 157 | $^{18}$F-FDG | TLG | 19.75% | 48.40% | 89.70% |
| Park [70]/2015 | NSCLC | 139 | $^{18}$F-FDG | MTV | 17.20% | 83.30% | 60.00% |

$^{18}$F-FDG: $^{18}$F-Fluoro-2-deoxy-d-glucose; $^{68}$Ga-FAPIs: gallium 68–labeled fibroblast-activation protein inhibitors; PET: positron emission tomography; SUVmax: Maximum standardized uptake value; MTV: Metabolic tumor volume; TLG: Total lesion glycolysis; HF: Heterogeneity factor; SUR: Standardized uptake ratio; NSCLC: non-small cell lung cancer.

#### 4.2.2. [F-18] FLT

[F-18] FLT (3′-deoxy-3′-fluorothymidine) is a stable PET tracer. $^{18}$FLT is slowly catabolised in vivo and retained in proliferating tissues following phosphorylation by thymidine kinase 1 (TK1) [110]. $^{18}$FLT is proportional to the proliferative activity of tumours. Researchers searched for selective cell proliferation imaging agents to overcome the decreased uptake of $^{18}$FDG after treatment [110]. Because $^{18}$F-FDG-PET showed false-positive uptake in areas of inflammation, Nakajo reported that $^{18}$F-FLT-PET showed greater specificity in staging lymph nodes in gastrointestinal tumors compared to $^{18}$F-FDG-PET [111]. Staniuk [112] evaluated local lymph node uptake in 22 patients with gastric cancer and identified one patient with micrometastasis. $^{18}$FLT-PET/CT examination confirmed a lymph node metastasis rate of 73%. The study concluded that $^{18}$FLT-PET/CT was an effective method to evaluate primary tumor

and local lymph node metastasis, and is useful and beneficial for the diagnosis and further treatment evaluation of gastric cancer. The results suggest that [18]FLT-PET/CT has the potential to facilitate treatment decision-making and reduce unnecessary dissection procedures. Other studies concluded that [18]FLT-PET/CT may have an equal or better diagnostic value than [18]F-FDG PET/CT in detecting primary and lymph node lesions in gastric cancer [113]. Hermann et al. [114] concluded that in gastric cancer patients with low or no uptake of [18]F-FDG, [18]F-PLT showed good sensitivity and may provide an early assessment of response to neoadjuvant therapy. Therefore, [18]F-FLT is a potential valuable tracer for gastric cancer [115]. However, image quality and proliferation rate calculations are compromised by the rapid degradation of TK1 in vivo. As a result, for imaging tumor proliferation, the need for a simple imaging method remains.

### 4.3. Radiomics

Although [18]F-FDG is a common diagnostic modality in current clinical practice, in some patients with mucinous adenocarcinoma and indolent cell carcinoma, [18]F-FDG PET/CT gives negative results due to the high mucus component and low tumor cell density [116]. Therefore, the detection rate of lymph node metastasis is low when [18]F-FDG is used. Although some studies have demonstrated the superiority of new imaging agents to [18]F-FDG, these can only be considered potential modalities due to the limited coverage and smaller sample size compared with [18]F-FDG studies. A modality with high accuracy still needs to be explored.

The field of medical image analysis has been growing rapidly in recent decades. Innovations in medical imaging technology are driving the field towards quantitative imaging and facilitating the development of automated and reproducible analysis methods. Radiomics is a novel field in medical imaging [117], and performance better than conventional staging systems has been demonstrated [117,118]. The technology relies on automated or semi-automated software in established computer imaging modalities to convert digital medical images into high-dimensional data for the quantitative analysis of medical images [119] and the creation of clinical models. It has shown great potential in predicting the biological behaviour, histological subtype classification, lymph node metastasis, diagnosis, treatment and prognosis of tumours [118,119]. Radiomics is capable of more reliably extracting the image features of lymph node metastasis in tomographic images (e.g., CT, MR, PET). Associating radiomic features with clinical features may improve the accuracy of lymph node metastasis detection in tumours [120]. Dong et al. [13] used a deep learning radiomics nomogram (DLRN) to study 730 patients with locally advanced gastric cancer. They found that 81.7% of OLNM without typical signs on CT could be detected by DLRN. This suggested that DLRN can complement CT and improve the accuracy of gastric cancer staging. Moreover, for patients with detected OLNM, individualized diagnosis and treatment plans were provided. Zhong et al. [121] used CT radiomics of primary tumors to predict cervical lymph node metastasis in tongue cancer by constructing artificial neural network-based models, and one of the models provided a reduction of OLNM from 30.9% to a minimum of 12.7%. Another study by Zhong et al. [122] examined 492 patients who did not receive preoperative enhanced CT based on multifunctional radiological features for the detection of OLNM. The conclusion was that radiomics could be used to predict OLNM in lung adenocarcinoma. The relevant data are shown in Table 4. The potential of radiomics for predicting OLNM in gastric cancer has also been demonstrated. CT-based radiomics as a non-invasive tool is expected to be useful for the individual prediction of lymph node metastasis in gastric cancer. In the future, the development of PET/CT radiomics can provide more accurate diagnoses of lymph node metastasis.

**Table 4.** Predicting OLNM in patients with different cancers using radiomics.

| Author/Year | Types of Cancer | No. Patients | Radiomics Method | AUC | Conclusion |
|---|---|---|---|---|---|
| Zhong [121]/2022 | Tongue cancer | 33 | ANN | 0.943 (Sensitivity: 93.10%; Specificity: 76.50%) | Using CT radiomics of the primary tumor, the rate of OLNM decreased from 30.9% to a minimum of 12.7% in the T1–2 group. |
| Dong [13]/2020 | Gastric cancer | 730 | DLRN | 0.821 | DLRN can detect 81.7% of OLNM patients. |
| Zhong [122]/2018 | Lung adenocarcinoma | 492 | Relief-based feature and support vector machine classification | 0.972 (Sensitivity: 94.80%; Specificity: 92.00%) | Radiomics predicts occult mediastinal LN metastases with 91.1% accuracy. |

OLNM: occult lymph node metastasis; AUC: area under curve; DLRN: Deep learning radiomic nomogram; ANN: Artificial neural network.

## 5. Discussion

In our study, we found that PET/CT is of substantial value for the diagnosis of OLNM. SUVmax is the most commonly used semi-quantitative parameter in clinical practice, but it reflects the metabolic activity of tumor cell components instead of the whole tumor and is susceptible to image resolution issues and noise. Therefore, other parameters reflecting tumor load and metabolic activity from the whole tumor lesion, such as MTV and TLG, are being evaluated. TLG and MTV can be factored in as complementary parameters to determine the presence of metastasis. The current development of new tumor tracers has helped to improve the detection rate of OLNM by PET/CT. Novel technologies [13,123] including a deep learning-based radiomic nomogram have been reported and have shown good predictive value for lymph node metastasis in locally advanced gastric cancer. Radiomics is also contributing significantly to a more accurate assessment of gastric cancer, and its value in OLNM detection in gastric cancer deserves further exploration.

With the development of new technologies to be used in combination with PET, the accuracy of OLNM detection is subject to great improvements. For example, multi-helical CT (MDCT) has the advantages of high resolution and easy operation, and advanced techniques such as multi-planar reconstruction or three-dimensional imaging are capable of detecting almost all positive lymph nodes [124]. The combination of PET/CT and MDCT has the potential to improve the accuracy, sensitivity, specificity, and possibly the diagnosis of lymph node metastasis. The combination of PET/Magnetic Resonance Imaging (MRI) can provide anatomical, functional and metabolic information together, which is potentially valuable for the identification of lymph node metastasis. In recurrent gastric cancer, the addition of PET/MRI to MDCT may also improve the diagnosis of lymph node metastasis.

As a large amount of data is required to build mathematical models for artificial intelligence analyses, the establishment of a multicentre database is important for disease prediction. It has been suggested that these machine learning models can be used to predict gastric cancer lymph node metastasis and benefit its staging and treatment. For example, faster region-based convolutional neural networks (FR-CNN) can contribute to reconstructing the distribution of lymph nodes in the body from multiple angles and provide an improved detection rate. The identification accuracy of FR-CNN for lymph node metastasis is 95.4%, which indicates its significant potential value in the diagnosis and treatment of gastric cancer [125]. Convolutional neural networks (CNNs) can automatically determine SUV based on images, and SUV determination, and when integrated into the PET data processing workflow will greatly facilitate the calculation of SUR, thereby reducing the clinical workload [99]. In the future, the detection rate of OLNM can be greatly improved under a precision medicine model, eventually leading a qualitative leap in the survival rate and quality of life of patients.

This article is subject to certain limitations. First, this is a retrospective review and [18]F-FDG is still a common clinical parameter for the diagnosis of lymph nodes and distant metastases, and, as mentioned before, [18]F-FDG remains extremely challenging for the detection of OLNM in gastric cancer. Data from studies with larger sample sizes are lacking to support the value of PET/CT in predicting OLNM in patients with gastric cancer. A large number of studies, such as those demonstrating the utility of various metabolic parameters in assessing patient response to treatment and prognostic evaluation, are needed to further investigate and confirm this. Although new imaging agents have shown some advantages, prospective studies are lacking. Additionally, although the noninvasive, quantified, and visualized features extracted by radiomics can reflect tumor heterogeneity, the extraction of [18]F-FDG PET/CT histological features is still strongly influenced by image acquisition and reconstruction parameters. Overall, various technical difficulties are yet to be overcome.

## 6. Conclusions

The latest trends in improving occult lymph node metastasis through non-invasive methods are reviewed. Currently, PET/CT still has many limitations for the detection of lymph node metastasis in gastric cancer. SUVmax of [18]F-FDG PET/CT is important and remains the most popular parameter for this purpose. Meanwhile, new imaging agents for PET are being developed with the aim of complementing the limitation of PET in that it only characterizes OLNM in terms of glucose metabolism, and novel technologies such as radiomics in synergy with artificial intelligence are advancing to provide more accurate and effective predictive tools.

**Author Contributions:** Conceptualization, J.W. and C.W.; methodology, C.W.; writing—original draft preparation, D.M.; data curation, D.M. and Y.Z.; writing—review and editing, X.S., Y.Z. and C.W.; project administration, J.W; funding acquisition, J.W. and C.W. All authors have read and agreed to the published version of the manuscript.

**Funding:** This work was supported by the Major Science and Technology Projects of Changzhou Health Care Commission, (No. ZD201901, to J.W.); Maternal and Child Health Association Foundation of Jiangsu, (No. FYX202017, to C.W.).

**Institutional Review Board Statement:** Not applicable.

**Informed Consent Statement:** Not applicable.

**Data Availability Statement:** No new data were created or analyzed in this study. Data sharing is not applicable to this article.

**Conflicts of Interest:** The authors declare no conflict of interest.

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
