# Peer review of "PET/CT for Predicting Occult Lymph Node Metastasis in Gastric Cancer"

_curroncol, doi:10.3390/curroncol29090513_

Round 1

Reviewer 1 Report

Summary

The submitted review article aims to explore the feasibility of PET/CT in detecting occult lymph node metastasis (OLNM) in gastric cancer and predict its effectiveness. The authors have reviewed eight metabolic parameters, including the common parameter SUVmax. The authors also propose to review the new PET imaging agents discussing future trends.

In section 1, FDG-based methods are described in six separate sections, including Table 1 as the summary of these metabolic parameters. Table 2 summarizes the characteristics of each parameter, including the deficiency of each parameter. In Section 2, the authors have discussed new imaging agents and reviewed the existing literature on other cancer types.

In the discussion section, the authors mentioned discussing the metabolic parameters of OLNM in all cancer patients and concluded that PET/CT scan for OLNM occupies the place. Authors conclude that SUR and TLG may be superior to other PET metabolic parameters.

Reviewer's Comments to Authors:

While authors claim to conclude that PET/CT scan of OLNM occupies the place, more constructive or substantial pieces of evidence must be presented for gastric cancer patients.

Authors also conclude that SUR and TLG may be superior to other metabolic parameters of PET. This claim has to be sufficiently supported by pieces of evidence and a thorough review.

Detecting lymph node metastasis in gastric cancer using 18F-FDG PET imaging remains controversial and has presented several challenges. While many studies reported these challenges, it would be essential to consider the following points and parameters when reviewing the feasibility of 18F-FDG PET/CT imaging in predicting occult lymph node metastasis.

-        Because primary tumor histology varies in gastric cancer and is often non-FDG avid, the metastasis of cells that do not show detectable uptake of FDG will not appear in 18F-FDG PET scan (reference PMID: 16228236). In this regard, a new PET tracer capable of detecting non-FDG-avid tumors and lymph nodes could be valuable.

-        Kim EY et al. 2011, reported that lymph node metastasis was not detectable in up to 70% of patients with non-FDG avid primary tumors. (PMID: 20226612). The SUVmax of the primary tumors is associated with that of the lymph nodes.

-        The metastatic lymph node size is a critical factor to consider when evaluating gastric cancer. Lymph node size is an important physiological parameter because the lymph node tumors can be as small as 3mm or even low. The spatial resolution limits of PET scanners are within 4 to 6 mm, and the feasibility of detecting lymph node metastasis via 18F-FDG PET remains challenging unless these limitations are overcome through technological developments. 

-        The high physiological uptake of 18F-FDG in the normal gastric wall gives the radioactive volume effects and background noise. Therefore the N staging of small lymph node metastasis remains a challenge. The first step would be demonstrating the feasibility of N staging in gastric cancer to address the ambiguities associated with lymph node metastasis, especially in gastric cancer, where N staging remains more challenging due to the involuntary movement of gastric walls. 

Strength:

It appears that 68Ga-FAPI is the only tracer evaluated in gastric cancer and shown to have higher diagnostic sensitivity and tumor to background contrast than 18F-FDG.

Weakness:

While authors refer that 68Ga-FAPI is better than 18F-FDG in detecting lymph node metastasis and could be helpful in the prediction of OLNM, justification should be provided on how this can be useful in predicting the OLNM in gastric cancer. 

The review is more descriptive and lacks a clear focus on OLNM in gastric cancer. The review often diverts from its stated title and objectives. Discussion is needed on how specific metabolic parameters and new tracers could be valuable in detecting the OLNM in gastric cancer.

The overall style and content need to be better organized and structured to present a meaningful analysis and address the aim of this review, as stated in the abstract and introduction sections.

It would be helpful first to introduce the occult lymph node metastasis (OLNM) and how the prediction of OLNM would be valuable to improving the gastric cancer outcome.

Authors must orient their discussion toward the tracer utility in detecting OLNM than the tumors. And how specific parameters, such as SUVmax, SUR, and TLG, could play a role in the prediction of OLNM.

While the review title is focused on gastric cancer, the literature reviewed and referenced here focuses on other cancer types. The review article must present the arguments based on the pieces of evidence available on gastric cancer and approaches to predicting OLNM based on PET imaging.

Authors often refer to PET/CT in the paragraph without mentioning which tracer they are referring to. It would be more appropriate to specify 18F-FDG PET/CT than only PET/CT. In the statements where the authors note SUV regarding specific tracers, please specify the tracer before saying SUV. 

It is challenging for a reader to follow through with the narrative and build on the information shared in the review. Grammar, style, and editing corrections are required to clarify the manuscript's writing.

The article needs to provide substantial evidence on OLNM in gastric cancer. Authors first must refer to the literature to discuss why OLNM is challenging to identify in gastric cancer, current gaps, and problems, and construct their arguments on OLNM in gastric cancer only.  

Specific comments:

11C-4DST – Was this tracer used in determining lymph node metastasis in gastric cancer patients?

18F-FLT- How does FLT correlate with OLNM? Were there any data reported regarding the FLT uptake in OLNM?

Line 39 – 45: OLNM is defined as N0 stage, which is not detectable by conventional palpation and imaging. - It is not clear what imaging modality is being referred to here. It would be appropriate first to discuss the characterization of OLNM in gastric cancer.

Line 46-47: What radiotracer are the authors referring here?

Author Response

Comments:

While authors claim to conclude that PET/CT scan of OLNM occupies the place, more constructive or substantial pieces of evidence must be presented for gastric cancer patients.

Author response:

We think this is an excellent suggestion. We have revised the text to address your concerns and hope that it is now clearer. Please see page 4 of manuscript, lines 66-74.

Authors also conclude that SUR and TLG may be superior to other metabolic parameters of PET. This claim has to be sufficiently supported by pieces of evidence and a thorough review.

Author response:

Thank you very much for your suggestion. We apologize for our inappropriate conclusions; there is insufficient evidence that SUR and TLG may be superior to other metabolic parameters of PET, and we have re-reviewed the relevant literature and have revised our conclusions in timely. Please see page 13 of manuscript, lines 225-232.

Detecting lymph node metastasis in gastric cancer using 18F-FDG PET imaging remains controversial and has presented several challenges. While many studies reported these challenges, it would be essential to consider the following points and parameters when reviewing the feasibility of 18F-FDG PET/CT imaging in predicting occult lymph node metastasis.

Because primary tumor histology varies in gastric cancer and is often non-FDG avid, the metastasis of cells that do not show detectable uptake of FDG will not appear in 18F-FDG PET scan (reference PMID: 16228236). In this regard, a new PET tracer capable of detecting non-FDG-avid tumors and lymph nodes could be valuable

Kim EY et al. 2011, reported that lymph node metastasis was not detectable in up to 70% of patients with non-FDG avid primary tumors. (PMID: 20226612). The SUVmax of the primary tumors is associated with that of the lymph nodes.

The metastatic lymph node size is a critical factor to consider when evaluating gastric cancer. Lymph node size is an import20226612ant physiological parameter because the lymph node tumors can be as small as 3mm or even low. The spatial resolution limits of PET scanners are within 4 to 6 mm, and the feasibility of detecting lymph node metastasis via 18F-FDG PET remains challenging unless these limitations are overcome through technological developments.

The high physiological uptake of 18F-FDG in the normal gastric wall gives the radioactive volume effects and background noise. Therefore the N staging of small lymph node metastasis remains a challenge. The first step would be demonstrating the feasibility of N staging in gastric cancer to address the ambiguities associated with lymph node metastasis, especially in gastric cancer, where N staging remains more challenging due to the involuntary movement of gastric walls.

Author response:

Thank you for pointing this out. The reviewer is correct, and we have added the suggested content to the revised manuscript. Please see page 5 of manuscript, lines 81-99.

While authors refer that 68Ga-FAPI is better than 18F-FDG in detecting lymph node metastasis and could be helpful in the prediction of OLNM, justification should be provided on how this can be useful in predicting the OLNM in gastric cancer. 

Author response:

We feel sorry that we did not provide enough information about 68Ga-FAPI can be useful in predicting the OLNM in gastric cancer. We have added relevant content in the revised manuscript. Please see page 14 of manuscript, lines 242-249.

The review is more descriptive and lacks a clear focus on OLNM in gastric cancer. The review often diverts from its stated title and objectives. Discussion is needed on how specific metabolic parameters and new tracers could be valuable in detecting the OLNM in gastric cancer.

Author response:

We feel great thanks for your professional review work on our article. As you are concerned, there are several problems that need to be addressed. According to your nice suggestions, we have made extensive corrections to our previous draft. We have added necessary literature to supplement our results and edited our article extensively and now hope that it is now clearer. The detailed corrections are listed below. Page 8, Lines 144-152, and page 9, lines 157-163, and page 14, lines 242-252. And we agree that this is a potential limitation of the study. We have added this as a limitation on page 21, lines 347-353, in the revised manuscript.

The overall style and content need to be better organized and structured to present a meaningful analysis and address the aim of this review, as stated in the abstract and introduction sections.

Author response:

Thanks for your suggestion. And we have changed it according to your suggestion.

It would be helpful first to introduce the occult lymph node metastasis (OLNM) and how the prediction of OLNM would be valuable to improving the gastric cancer outcome.

Author response:

As suggested by the reviewer, we have added to the literature related to the detection of OLNM to improve the prognosis of patients with gastric cancer. in the revised manuscript. Please see page 4 of manuscript, Lines 66-72.

Authors must orient their discussion toward the tracer utility in detecting OLNM than the tumors. And how specific parameters, such as SUVmax, SUR, and TLG, could play a role in the prediction of OLNM.

Author response:

We feel great thanks for your professional review work on our article. As you are concerned, there are several problems that need to be addressed. According to your nice suggestions, we have made extensive corrections to our previous draft. We have added necessary literature to supplement our results and edited our article extensively and now hope that it is now clearer. The detailed corrections are listed below. Page 8, Lines 144-152, and page 9, lines 157-163, and page 10, lines 184-189, and page 12, lines 195-200, and page 13, lines 211-216. And we agree that this is a potential limitation of the study. We have added this as a limitation on page 21, lines 347-353, in the revised manuscript.

While the review title is focused on gastric cancer, the literature reviewed and referenced here focuses on other cancer types. The review article must present the arguments based on the pieces of evidence available on gastric cancer and approaches to predicting OLNM based on PET imaging.

Author response:

We regret that we have not provided enough relevant information and we have added relevant literature in our revised manuscript. Please see page 4 of manuscript, Lines 66-72. And we agree that this is a potential limitation of the study. We have added this as a limitation on page 21, lines 347-353, in the revised manuscript.

Authors often refer to PET/CT in the paragraph without mentioning which tracer they are referring to. It would be more appropriate to specify 18F-FDG PET/CT than only PET/CT. In the statements where the authors note SUV regarding specific tracers, please specify the tracer before saying SUV. 

Author response:

Thank you for your suggestion. We feel really sorry for our carelessness and we've changed the title "FDG-based methods" to "18FDG-based methods" (page7, line124).

It is challenging for a reader to follow through with the narrative and build on the information shared in the review. Grammar, style, and editing corrections are required to clarify the manuscript's writing.

Author response:

Thanks for your suggestions. We feel sorry for our poor writings, however, we do invite a friend of us who is a native English speaker from USA help polish our article. Due to our friend’s help, the article was edited extensively. And we hope the revised manuscript could be acceptable for you.

The article needs to provide substantial evidence on OLNM in gastric cancer. Authors first must refer to the literature to discuss why OLNM is challenging to identify in gastric cancer, current gaps, and problems, and construct their arguments on OLNM in gastric cancer only. 

Author response:

We feel great thanks for your professional review work on our article. As you are concerned, there are several problems that need to be addressed. According to your nice suggestions, we have made extensive corrections to our previous draft. We have added necessary data to supplement our results and edited our article extensively. The detailed corrections in the revised manuscript. Please see page 4 of manuscript, lines 58-60, and page 5, lines81-110.

11C-4DST – Was this tracer used in determining lymph node metastasis in gastric cancer patients?

Author response:

Thank you for pointing this out. We think this is an excellent suggestion. We have carefully reviewed the content in question and have removed it as we felt it did not fit the topic of the article.

18F-FLT- How does FLT correlate with OLNM? Were there any data reported regarding the FLT uptake in OLNM?

Author response:

Thank you for your suggestion. However, there is no direct literature on 18F-FLT with OLNM in our study, but we found its potential value in studying lymph node metastasis in gastric cancer. Please see page 16 of manuscript, lines 265-273.

Line 39 – 45: OLNM is defined as N0 stage, which is not detectable by conventional palpation and imaging. It is not clear what imaging modality is being referred to here. It would be appropriate first to discuss the characterization of OLNM in gastric cancer.

Author response:

CT is a common imaging modality used in clinical practice. Here, it refers to lymph node metastases without typical signs on CT or 18F-FDG PET/CT, which are eventually pathologically confirmed. Please see page 4 of manuscript, lines 58-60, and lines 68-72.

Line 46-47: What radiotracer are the authors referring here?

Author response:

Thank you for your suggestion, it is 18F-FDG and we have modified it.

Reviewer 2 Report

The authors made a good job of reviewing the literature but it would be better to add the criteria used to perform it. 

I suggest modifying the Metabolic Total Volume into Metabolic Tumor Volume

(Role of metabolic tumor volume (MTV) and standardized uptake value (SUV) based parameters derived from whole body (WB) 18F-FDG PET/CT in interim treatment response assessment of NSCLC

Akshima Sharma, Anant Mohan, Ashu Bhalla, Sreenivas VISHNUBHATLA, Anil Pandey, GEETANJALI Arora, Sanjay Thulkar, Chandan Das, Chetan Patel, Chandrasekhar Bal and Rakesh Kumar

Journal of Nuclear Medicine May 2019, 60 (supplement 1) 1326;)

It might be effective to visualize how many works are in the literature for the keywords: 18FDG PET-CT, gastric cancer and occult lymph node metastasis, and so on.

Please modify:
Line 87 to 89 Some studies point out that SUVmax is a predictor of OLNM, the greater the SUVmax, the higher the incidence of lymph node metastasis(30).  

into

Some studies point out that the SUVmax of the primary tumor is a predictor of OLNM, the greater the SUVmax, the higher the incidence of lymph node metastasis(30).

Author Response

1. The authors made a good job of reviewing the literature but it would be better to add the criteria used to perform it.

3. It might be effective to visualize how many works are in the literature for the keywords: 18FDG PET-CT, gastric cancer and occult lymph node metastasis, and so on.

The author responds: Thanks for your suggestion. We added performance criteria. See page 6 of the manuscript, lines 111-122.

2. I suggest modifying the Metabolic Total Volume into Metabolic Tumor Volume

The author responds:

This is indeed a huge mistake for the overall quality of our article. We feel sorry for our carelessness. We have changed the total metabolic volume to the metabolic tumor volume, and we appreciate your pointing out.

Reviewer 3 Report

The paper is interesting and subject is well developed.

Some suggestions/corrections:

1) Ln 44: sentence “OLNM is an important prognostic factor” is repetitive; it should be deleted; 2) Ln 86: in sentence “It is used to assess the response status of lymph nodes..” I would add “for example”;

3) I encourage authors to add one table with general characteristics of included studies (for example author, year, kind of studied cancers population, number of patients, PET tracer used, PET parameter evaluated, sensitivity and specificity in detecting/predicting OLNM  

Author Response

Response to reviewer 3 comments

Comments:

1) Ln 44: sentence “OLNM is an important prognostic factor” is repetitive; it should be deleted; 

Author response:

According to your suggestion, we have corrected it.

2) Ln 86: in sentence “It is used to assess the response status of lymph nodes.” I would add “for example”;

Author response:

Thanks for your suggestions. We feel sorry for the improper wording. We have added the “for example” as you suggested.

3) I encourage authors to add one table with general characteristics of included studies (for example author, year, kind of studied cancers population, number of patients, PET tracer used, PET parameter evaluated, sensitivity and specificity in detecting/predicting OLNM

Author response:

Thank you for pointing this out. The table was embedded in the paper. Please see page 15 of the revised manuscript.

Reviewer 4 Report

The study assesses a current, timely topic in gastric cancer. 
We recommend some changes:
- We believe this article is suitable for publication in the journal although major revisions are needed. The main strengths of this paper are that it addresses an interesting and very timely question and provides a clear answer, with some limitations. 

- A table summarizing the main results reported in the Radiomics chapter should be included. Similarly, we believe at least two figures should be included in order to summarize some findings. In fact, the paper appears quite "poor".

- The background of the changing scenario of medical treatment in gastric cancer should be better discussed, and some recent papers regarding this topic should be included ( PMID: 33916915 ; PMID: 33916206; PMID: 31793342)

- A linguistic revision is needed.
Major changes are necessary.

Author Response

Response to reviewer 4 comments

Comments:

- A table summarizing the main results reported in the Radiomics chapter should be included. Similarly, we believe at least two figures should be included in order to summarize some findings.

Author response:

Thank you for pointing this out. We agree that this is an important consideration. These tables are embedded in the paper. Please see page 15 and 18 of the revised manuscript.

- The background of the changing scenario of medical treatment in gastric cancer should be better discussed, and some recent papers regarding this topic should be included ( PMID: 33916915 ; PMID: 33916206; PMID: 31793342)

Author response:

Thank you for pointing this out. Although we agree that this is an important consideration, it is beyond the scope of this manuscript.

Round 2

Reviewer 1 Report

Line 45-84: The traditional criteria for 18F-FDG PET/CT diagnosis of lymph node metastasis in gastric cancer are based on the lymph nodes themselves, with a short diameter of ≥10 mm (15) on CT images or lymph node uptake of 18F-FDG similar to or higher than the liver uptake of 18F-FDG on PET images (16).

Comment: This statement is confusing and needs rewording into two separate statements for CT and PET criteria.

Line 49-53: More than 60% of metastatic lymph nodes in gastric cancer are less than 8 mm in size(17), and the spatial and temporal specificity of the lesion tissue may also lead to a reduced detection rate of metastatic lymph nodes on CT. It has been suggested that 18F-FDG PET/CT uses glucose metabolism rather than changes in lymph node size to diagnose lymph node metastasis(11, 18).

Comment: This statement is confusing and can be re-written to distinguish between reduced detection of the lymph node on CT due to small size and How 18F-FDG PET could be helpful here.

Line 87-99: During the first review, authors were recommended to make changes by integrating the missing aspect of 18F-FDG PET imaging in gastric cancer. The reviewer advised few key points to be included in this review article. However, these were included as statements in this revised manuscript. These comments aimed to orient the discussion and help authors elaborate on these factors that potentially impact the detection of OLNM in gastric cancer.

Comment: Authors are advised to integrate these factors with elaborative discussions in the manuscript by organizing them in respective sections. 18F-FDG PET imaging advantages, limitations, and potential approaches to detect OLNM in gastric cancer can be discussed in the subsection. For example, these can be covered as follows:

·       18F-FDG PET advantages in gastric cancer

·       18F-FDG PET limitation and challenges in gastric cancer

·       Potential approaches for the detection of OLNM in gastric cancer

Overall comment:

The writing needs improvements. It is still challenging for a reader to follow through with arguments made or the narrative shared in the written manuscript. The review article needs significant revision in English language, formatting, and narratives. The text often diverts abruptly from the title and objective of discussing the feasibility of detecting OLNM using PET/CT in gastric cancer.

Significant work is still needed to correct grammar, improve the narration style, and edit. Authors must get help from writing services. 

Author Response

Response to the comments from Reviewer 1:

Line 45-84: The traditional criteria for 18F-FDG PET/CT diagnosis of lymph node metastasis in gastric cancer are based on the lymph nodes themselves, with a short diameter of ≥10 mm (15) on CT images or lymph node uptake of 18F-FDG similar to or higher than the liver uptake of 18F-FDG on PET images (16).

Comment: This statement is confusing and needs rewording into two separate statements for CT and PET criteria.

 Author response: Thank you for your advice. We have divided the criteria for CT and PET into two parts. Please see page 5, lines 79-80 and lines 88-89, in the revised manuscript.

Line 49-53: More than 60% of metastatic lymph nodes in gastric cancer are less than 8 mm in size(17), and the spatial and temporal specificity of the lesion tissue may also lead to a reduced detection rate of metastatic lymph nodes on CT. It has been suggested that 18F-FDG PET/CT uses glucose metabolism rather than changes in lymph node size to diagnose lymph node metastasis (11, 18).

Comment: This statement is confusing and can be re-written to distinguish between reduced detection of the lymph node on CT due to small size and How 18F-FDG PET could be helpful here.

 Author response: We are very grateful to your comments on the manuscript. We have re-written this section and hope it can express clearly. Please see page 5, lines 81-87, in the revised manuscript.

Line 87-99: During the first review, authors were recommended to make changes by integrating the missing aspect of 18F-FDG PET imaging in gastric cancer. The reviewer advised few key points to be included in this review article. However, these were included as statements in this revised manuscript. These comments aimed to orient the discussion and help authors elaborate on these factors that potentially impact the detection of OLNM in gastric cancer.

Comment: Authors are advised to integrate these factors with elaborative discussions in the manuscript by organizing them in respective sections. 18F-FDG PET imaging advantages, limitations, and potential approaches to detect OLNM in gastric cancer can be discussed in the subsection. For example, these can be covered as follows:

  • 18F-FDG PET advantages in gastric cancer
  • 18F-FDG PET limitation and challenges in gastric cancer
  • Potential approaches for the detection of OLNM in gastric cancer

Author response: We feel great thanks for your professional review work on our article. According to your nice suggestions, we have made extensive corrections to our previous draft. We have revised our section headings, as follows:

1, Advantages of 18F-FDG PET in detecting lymph node metastasis in gastric cancer

2, The limitations and challenges of 18F-FDG PET in gastric cancer

3, Common methods for detecting OLNM based on 18FDG

4, Potential approaches for the detection of OLNM in gastric cancer

 (18FDG-based methods; Novel imaging agents; radiomics)

I hope that the changes I have made resolve all your concerns about the article.

Overall comment:

The writing needs improvements. It is still challenging for a reader to follow through with arguments made or the narrative shared in the written manuscript. The review article needs significant revision in English language, formatting, and narratives. The text often diverts abruptly from the title and objective of discussing the feasibility of detecting OLNM using PET/CT in gastric cancer.

Significant work is still needed to correct grammar, improve the narration style, and edit. Authors must get help from writing services. 

Author response: Thanks for your suggestion. We feel sorry for our poor writings. We have tried our best to polish the language in the revised manuscript. We hope the revised manuscript could be acceptable for you.